# Comparison of FDA accelerated vs regular pathway approvals for lung cancer treatments between 2006 and 2018

**Tatiane Bomfim Ribeiro**[1]*, **Lewis Buss**[1], **Cole Wayant**[2], **Moacyr Roberto Cuce Nobre**[1,3]

**1** Departamento de Medicina Preventiva, Faculdade de Medicina, Universidade de São Paulo, São Paulo, SP, Brazil, **2** Department of Biomedical Sciences, Oklahoma State University Center for Health Science, Tulsa, OK, United States of America, **3** InCor, Instituto o Coração, Hospital das Clínicas HCFMUSP, Faculdade de Medicina, Universidade de Sao Paulo, Sao Paulo, SP, Brazil

* tatianeribeiro6@gmail.com

**Data Availability Statement:** All relevant data are within the manuscript and its Supporting Information files.

## Abstract

Regulatory agencies around the world have been using flexible requirements for approval of new drugs, especially for cancer drugs. The US Food and Drug Administration (FDA) is mostly the first agency to approve new drugs worldwide, mainly due to the faster terms of the accelerated pathway and breakthrough therapy designation. Surrogate endpoints and preliminary data (e.g. single-arm and phase 2 studies) are used for these new approvals, however larger effect sizes are expected. **We aim to** compare FDA Accelerated vs Regular Pathway approvals and Breakthrough therapy designations (BTD) for lung cancer treatments between 2006 and 2018 regarding study design, sample size, outcome measures and effect size. **We assessed the FDA database to collect data from** studies that formed the basis of approvals of new drugs or indications for lung cancer spanning from 2006 to 2018. **We found that** accelerated pathway approvals are based on significantly more single-arm studies with small sample sizes and surrogate primary endpoints. However, effect size was not different between the pathways. A large proportion of studies used to support regular pathway approvals also showed these characteristics that are related to low quality and uncertain evidence. Compared to other approvals, BTD were more frequently based on single-arm studies. There was no significant difference in use of surrogate endpoints or sample size. 44% of BTD were based on studies demonstrating large effect sizes, proportionally more than approvals not receiving this designation. **In conclusion,** based on the indicators of evidence quality we extracted, criteria's for granting accelerated approval and breakthrough therapy designation seen not clear. Faster approvals are in the majority full of uncertainties which should be viewed with caution and the patient have to be communicated to allow shared decision making. Post-marketing validation is essential.

## Introduction

For a new drug to enter the market, a local health agency must assess the efficacy and safety based on studies submitted by pharmaceutical companies. The North American agency, the

**Funding:** CW is supported by the National Cancer Institute of the National Institutes of Health under award number F30CA243651-01. No additional external funding was received for this study.

**Competing interests:** The authors have declared that no competing interests exist.

US Food and Drug Administration (FDA), is the in the great majority the first agency in the world to approve new drugs [1].

Cancer is the leading cause of death in high-income countries [2]. Lung cancer incidence is increasing worldwide, with over two million new cases in 2018 [3]. The treatment arsenal for lung cancer is extensive, with an improved understanding of the genetic drivers (e.g. EGFR, ALK, BRAF) leading to a rapid growth in novel medications. Targeted and immune therapies are first-line treatments and widely recommended in current guidelines [4–6].

Considering recent FDA approvals, protein kinase inhibitors for advanced lung cancer accounted for 53% of new therapies for the four most common neoplasms and 78% were granted a special designation for faster approval [7]. Other studies have shown that these innovative trials of new FDA-approved anti-cancer drugs focus heavily on surrogate markers of clinical benefit, and trials that measure overall survival demonstrate only modest improvements [8].

In 1992, during the HIV crisis, the FDA created the "Expedited Programs for Serious Conditions". The aim was to provide faster access to medications that addressed 'unmet medical need' though an accelerated approval pathway and priority review designation in which studies using surrogate endpoints were considered. Subsequently other designations have been created to increase the flexibility of the approvals process. Under the "Breakthrough" designation, implemented in 2012, approvals can be based on preliminary clinical evidence–for example phase 2 or single-arm studies–that "indicates the drug may demonstrate substantial improvement over available therapy on a clinically significant endpoint(s)" [9]. While double blind randomized controlled trials (RCT) are the gold standard for intervention studies [10], these expedited programs result in more drug approvals based on single arm trials, with small sample sizes and surrogate endpoints [11–12].

Accelerated pathway differently from regular pathway allows anticancer drugs to be approved based on a surrogate endpoint, however it is unclear whether there is a difference in 1) proportion of surrogate endpoints, 2) sample size, 3) study design and, 4) effect size. Our two main comparisons were trials that received accelerated approval vs. regular approval and trials that received breakthrough designation vs. those that did not.

## Methods

This study used publicly available data and did not involve individual patient information. Ethics approval was not required. This report follows the STROBE statement for observational data reporting.

### Data collection

Two authors (TR and LB) independently reviewed the FDA's online bulletin (@DrugsFDA) that includes all approvals for lung cancer. The FDA database contains approval data on new drugs and indications since 2006. We retrieved information on approvals and trials supporting these approvals from January 1, 2006 to December 31, 2018. We excluded the following approval types: agnostic therapies, treatments for neuroendocrine tumors, changes to the dosing regimen of previously approved drugs or further specification of mutations (e.g. L858R), and finally we excluded conversion from accelerated to regular in order to due to duplicate the approval for the specific condition. First approvals of new lung cancer (NSCLC and SCLC) drugs as well as novel indications were included. One approval was not included in the FDA material but identified after cross-referencing. The following data were collected about studies forming the basis of approvals: sample size, study type (open-label RCT, double-blind RCT or single-arm), the intervention and control, whether overall survival was measured as an

endpoint and the primary endpoint used to justify the FDA approval (overall survival—OS, progressive-free survival—PFS, overall response rate—OvRR or objective response rate—ORR), and the measure of effect. Generally, according to RECIST, OvRR is related to partial or complete response. ORR is defined as the sum of complete and partial responses [13]. However, the definition varies among studies and the authors often did not specify which definition they used.

We recorded the approval pathway (regular vs accelerated) and special designations (breakthrough or priority review). These data were subsequently confirmed on the accelerated approval (https://www.fda.gov/drugs/nda-and-bla-approvals/accelerated-approvals) and breakthrough therapy designation (BTD) databases (https://www.fda.gov/drugs/nda-and-bla-approvals/breakthrough-therapy-approvals). We checked if the drug was a new molecular entity (NME) approved for the first time in the USA by searching the generic names on the @Drugs FDA database.

## Magnitude of effect

We assessed the magnitude of effect for the specified primary outcome. According to the GRADE classification [14] a relative risk (RR) greater than or equal to two was considered "large" and those less than two were considered "not large"; and a hazard ratio (HR) based in outcomes as survival and progressive-free survival, greater than or less than .5 was considered "large" and those greater than .5 were considered "not large". For single-arm studies, which by definition included a lack a control group, we selected a historical control ("best control available") to estimate the magnitude of effect (Fig 1). If the drug had a biomarker (EGFR, ALK, BRAF or ROS-1) and there was a previous FDA approved drug for the same indication and treatment line, then this drug was selected as the control. However, for drugs without a biomarker or a previous FDA approved comparator we used a historical control from the "Major Milestones against Cancer Timeline for Lung cancer" [15], available at the ASCO website. Additionally, when necessary we consulted the NCCN Guidelines for Non-small lung cancer [16] and Small Cell lung cancer [17].

## Statistical analysis

The main variables were grouped into the following binary categories: RCT (blinded and open-label) vs single-arm; surrogate outcomes (OvRR, ORR and PFS) vs overall survival; sample sizes of $\geq$200 vs <200 (following Ladanie et al., 2019 [18] recommendations); and magnitude of effect as large vs not large. Study characteristics were compared between accelerated

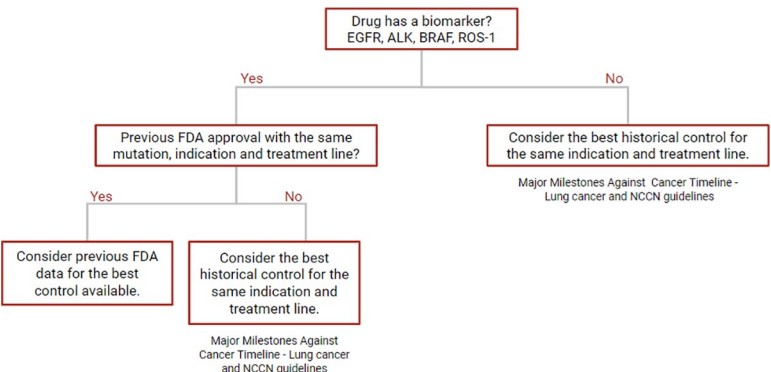

**Fig 1. Selection of historical controls to estimate effect sizes in single-arms studies.**

and regular approvals and this analysis was repeated for BTD for approvals from 2012 onwards, comparing BTD vs non-BTD, the absolute number and percentage was presented. We plotted effect sizes from Accelerated Approval Pathway and BTD according to its effect size measurement; one graph for studies based in hazard ratio (time to event outcome: OS and PFS) and another graph to studies based in RR estimated from single-arms studies using response rate as the primary endpoint.

The study was primarily descriptive, and we included all data available for lung cancer treatment in the period (2006–2018). The absolute difference (AD) and prevalence ratio (PR) with a confidence interval of 95% based on a Normal approximation of the sampling distribution were calculated. A 2-sided *P* values .05 was used to assess statistical significance by the Fisher's exact test. Analyses were performed in R (R Project for Statistical Computing) version 3.5.3.

## Results

### Regulatory characteristics

From 2006 to 2018, the FDA granted 33 approvals for new drugs and supplemental indications in lung cancer (Table 1). The approvals excluded and justification are available at Supporting Information (S1 Table). 33% (11 of 33) of drugs approved via the accelerated pathway and 42% (14 of 33) received BTD. Priority review designation was granted in 42% (14 of 33) of approvals, concomitantly with other designations in some cases. 30% (10 of 33) of approvals were NEMs.

**Study characteristics.** Thirty-seven studies formed the basis of the 33 approvals (Table 2). Single-arms studies and open-label RCTs accounted for 43% (16 of 37), with blinded RCTs for an additional 14% (5 of 37). OS was reported in 46% (17 of 37) of the studies and was specified as the primary endpoint in 30% (11 of 37). PFS, ORR and OvRR were the primary endpoint in 27% (10 of 37), 30% (11 of 37) and 13% (5 of 37) of studies, respectively.

### Elements related to the quality of evidence in lung cancer approvals

**Accelerated vs regular pathway.** The single-arm trials and small sample sized were observed 4.06 and 2.37 more times in the Accelerated pathway compared to the Regular pathway, with an absolute difference of 64% and 40%, respectively (Table 3). However, the number of trials with larger effect size was not different among the pathways.

Accelerated pathway are 1.58 (CI95% 1.09–2.30) more likely to have studies with surrogate endpoint and an absolute different between the groups of 37% was observed, however p-value was in the limit of the significance, according to Fisher's exact test, probably due to small sample size.

Graphs 1 and 2 shows that most studies approved using the Accelerated pathway were based in relative risk calculation (from single-arm studies) and only 5 studies showed a large effect size (Graph 2).

**Breakthrough therapy designation.** Thirty-two approvals were made between 2012 and 2018, of which 16 were granted BTD (Table 3). Single-arms trials were 2.5 more frequent in the group which received the BTD, with an absolute difference of 37%, however this difference was not significant according to Fisher's exact test.

Large effect sizes, surrogate endpoints and small sample sizes were not more common in BTD, with absolute difference of 31%, 19% and 0%, respectively.

Graph 3 and 4 showed that several studies since 2012 received BTD. The majority of the studies which received this designation were based on relative risk calculation (from single-arm studies) (Graph 4), as expected, larger effect sizes were noted in these studies. Details on

**Table 1. Regulatory characteristics of FDA approvals for lung cancer from 2006 to 2018.**

| Drug name (generic) | Brand name | Approval date | NME | Approval pathway | Designation 1 | Designation 2 |
|---|---|---|---|---|---|---|
| Bevacizumab | AVASTIN | Oct-06 | No | Regular | Not mentioned | Not mentioned |
| Pemetrexed | ALIMTA | Sep-08 | No | Accelerated | Not mentioned | Not mentioned |
| Erlotinib | TARCEVA | Apr-10 | No | Regular | Not mentioned | Not mentioned |
| Crizotinib | XALKORI | Aug-11 | Yes | Accelerated | Not mentioned | Not mentioned |
| Erlotinib | TARCEVA | May-13 | No | Regular | Not mentioned | Not mentioned |
| Afatinib | GILOTRIF | Jul-13 | Yes | Regular | Not mentioned | Not mentioned |
| Ceritinib | ZYKADIA | Apr-14 | Yes | Accelerated | Breakthrough therapy | Not mentioned |
| Ramucirumab | CYRAMZA | Dec-14 | No | Regular | Not mentioned | Not mentioned |
| Nivolumab | OPDIVO | Mar-15 | No | Regular | Not mentioned | Not mentioned |
| Gefitinib | IRESSA | Jul-15 | No | Regular | Not mentioned | Not mentioned |
| Pembrolizumab | KEYTRUDA | Oct-15 | No | Accelerated | Breakthrough therapy | Not mentioned |
| Nivolumab | OPDIVO | Oct-15 | No | Regular | Breakthrough therapy | Not mentioned |
| Osimertinib | TAGRISSO | Nov-15 | Yes | Accelerated | Breakthrough therapy | Priority review |
| Necitumumab | PORTAZZA | Nov-15 | Yes | Regular | Not mentioned | Not mentioned |
| Alectinib | ALECENSA | Dec-15 | Yes | Accelerated | Breakthrough therapy | Priority review |
| Crizotinib | XALKORI | Mar-16 | No | Regular | Breakthrough therapy | Priority review |
| Afatinib | GILOTRIF | Apr-16 | No | Regular | Not mentioned | Not mentioned |
| Atezolizumab | TECENTRIQ | Oct-16 | No | Regular | Breakthrough therapy | Not mentioned |
| Pembrolizumab | KEYTRUDA | Oct-16 | No | Regular | Breakthrough therapy | Priority review |
| Brigatinib | ALUNBRIG | Apr-17 | Yes | Accelerated | Breakthrough therapy | Priority review |
| Pembrolizumab | KEYTRUDA | May-17 | No | Accelerated | Priority review | Not mentioned |
| Dabrafenib and trametinib | TAFINLAR and MEKINIST | Jun-17 | No | Regular | Breakthrough therapy | Not mentioned |
| Afatinib | GILOTRIF | Jan-18 | No | Regular | Priority review | Not mentioned |
| Durvalumab | IMFINZI | Feb-18 | No | Regular | Breakthrough therapy | Priority review |
| Osimertinib | TAGRISSO | Apr-18 | No | Regular | Breakthrough therapy | Priority review |
| Nivolumab | OPDIVO | Aug-18 | No | Accelerated | Priority review | Not mentioned |
| Dacomitinib | VIZIMPRO | Sep-18 | Yes | Regular | Priority review | Not mentioned |
| Pembrolizumab | KEYTRUDA | Oct-18 | No | Regular | Priority review | Not mentioned |
| Lorlatinib | LORBRENA | Nov-18 | Yes | Accelerated | Breakthrough therapy | Priority review |
| Atezolizumab | TECENTRIQ | Dec-18 | No | Regular | Not mentioned | Not mentioned |

NME: new molecular entity (1st FDA approval).

the historical controls and RR calculations are presented in the Supporting Information—S2 Table.

## Discussion

We assessed characteristics of the evidence base for FDA lung cancer approvals between 2006 and 2018. Accelerated pathway approvals were 4.06 times more based on single-arm studies, studies with small sample sizes and surrogate primary endpoints. These results are expected given the purpose of accelerated approval is to provide the public more rapid access to new therapies. However, there was no difference in effect sizes between accelerated and regular approvals, likely because regular approvals used surrogate endpoints more often than overall survival. Such a finding is concerning because overall survival is the main patient-centered endpoint. It has previously been shown that surrogate endpoints tend to be associated with larger effect sizes compared to overall survival [19]. A recent review addressed the correlation

**Table 2. Characteristics of studies of lung cancer approvals by the FDA from 2006 to 2018.**

| Drug name (generic) | Approval date | N. st | Study design | Intervention arm | Control arm | Sample size | OS reported? | Primary endpoint | Effect size calculated (95%CI) | Magnitude of effect |
|---|---|---|---|---|---|---|---|---|---|---|
| Bevacizumab | Oct-06 | 1 | RCT open-label | bevacizumab+ carboplatin and paclitaxel | carboplatin and paclitaxel | 878 | Yes | OS | HR: 0.80 (NM) | Not large |
| Pemetrexed | Sep-08 | 1 | RCT open-label | pemetrexed + cisplatin | gemcitabine + cisplatin | 1725 | Yes | OS | HRa: 0.94 (0.84–1.05) | Not large [a] |
| Erlotinib | Apr-10 | 1 | RCT double-blind | erlotinib | placebo orally | 889 | Yes | PFS | HR: 0.71 (0.62–0.82) | Not large |
| Crizotinib | Aug-11 | 2.1 | A) Single-arm | crizotinib | no control | 136 | NM | ORR | RRc: 2.64 (2.15–3.24) | Large |
| Crizotinib | Aug-11 | 2.2 | B) Single-arm | crizotinib | no control | 119 | NM | ORR | RRc: 3.22 (2.67–3.88) | Large |
| Erlotinib | May-13 | 1 | RCT open-label | erlotinib | platinum-based doublet chemotherapy | 174 | Yes | PFS | HR: 0.34 (0.23–0.49) | Large |
| Afatinib | Jul-13 | 1 | RCT open-label | afatinib | pemetrexed/ cisplatin | 345 | NM | PFS | HR: 0.58 (0.43–0.78) | Not large |
| Ceritinib | Apr-14 | 1 | Single-arm | certinib | no control | 163 | NM | ORR | RRc: 0.68 (0.55–0.83) | Not large |
| Ramucirumab | Dec-14 | 1 | RCT double-blind | ramucirumab+docetaxel | placebo+docetaxel | 1253 | Yes | OS | HR: 0.86 (0.75–0.98) | Not large |
| Nivolumab | Mar-15 | 2.1 | A) RCT open-label | nivolumab | docetaxel | 272 | Yes | OS | HR: 0.59 (0.44–0.79) | Not large |
| Nivolumab | Mar-15 | 2.2 | B) Single-arm | nivolumab | NA | 117 | NM | ORR | RRc: 1.71 (0.86–3.42) | Not large [a] |
| Gefitinib | Jul-15 | 2.1 | A) Single-arm | gefitinib | no control | 106 | NM | ORR | RRc: 2.64 (2.11–3.30) | Large |
| Gefitinib | Jul-15 | 2.2 | B) RCT open-label | gefitinib | carboplatin/ paclitaxel | 186 | NM | PFS exA | HR: 0.54 (0.38–0.79) | Not large |
| Pembrolizumab | Oct-15 | 1 | Single-arm | pembrolizumab | no control | 61 | NM | ORR | RRc: 7.04 (3.06–16.19) | Large |
| Nivolumab | Oct-15 | 1 | RCT open-label | nivolumab | docetaxel | 582 | Yes | OS | HR: 0.73 (0.60–0.89) | Not large |
| Osimertinib | Nov-15 | 2.1 | A) Single-arm | osimertinib | no control | 201 | NM | ORR | RRc: 6.47 (4.75–8.82) | Large |
| Osimertinib | Nov-15 | 2.2 | B) Single-arm | osimertinib | no control | 210 | NM | ORR | RRc: 6.92 (5.10–9.39) | Large |
| Necitumumab | Nov-15 | 1 | RCT open-label | necitumumab+gemcitabine and cisplatin | gemcitabine and cisplatin (alone) | 1093 | Yes | OS | HR 0.84 (0.74–0.96) | Not large |

(*Continued*)

**Table 2.** (Continued)

| Drug name (generic) | Approval date | N. st | Study design | Intervention arm | Control arm | Sample size | OS reported? | Primary endpoint | Effect size calculated (95%CI) | Magnitude of effect |
|---|---|---|---|---|---|---|---|---|---|---|
| Alectinib | Dec-15 | 2.1 | A) Single-arm | alectinib | no control | 87 | NM | ORR | RRc: 0.86 (0.63–1.18) | Not large [a] |
| Alectinib | Dec-15 | 2.2 | B) Single-arm | alectinib | no control | 138 | NM | ORR | RRc: 1.00 (0.77–1.29) | Not large [a] |
| Crizotinib | Mar-16 | 1 | Single-arm | crizotinib | no control | 50 | NM | ORR | RRc: 3.48 (2.76–4.39) | Large |
| Afatinib | Apr-16 | 1 | RCT open-label | afatinib | Erlotinib | 795 | Yes | PFS | HR: 0.82 (0.68–0.99) | Not large |
| Atezolizumab | Oct-16 | 2.1 | A)RCT open-label | atezolizumab | docetaxel | 1137 (Both studies A +B) | Yes | OS | HR: 0.74 (0.63–0.87) | Not large |
| Atezolizumab | Oct-16 | 2.2 | B) RCT open-label | atezolizumab | docetaxel | 1137 (Both studies A +B) | Yes | OS | HR: 0.69 (0.52–0.92) | Not large |
| Pembrolizumab | Oct-16 | 1 | RCT open-label | pembrolizumab | platinum-based chemotherapy | 305 | Yes | PFS | HR: 0.50 (0.37–0.68) | Large |
| Brigatinib | Apr-17 | 1 | Single-arm | brigatinib | no control | 110 | NM | OvRR | RRc: 1.27 (0.94–1.17) | Not large [a] |
| Pembrolizumab | May-17 | 1 | RCT open-label | pembrolizumab +pemetrexed and carboplatin | pemetrexed and carboplatin | 123 | NM | PFS | HR: 0.53 (0.31–0.91) | Not large |
| Dabrafenib and trametinib | Jun-17 | 1 | Single-arm | dabrafenib and trametinib | no control | 36 | NM | OvRR | RRc: 3.22 (2.72–3.81) | Large |
| Afatinib | Jan-18 | 1 | Single-arm | afatinib | no control | 32 (total of 3 studies) | NM | OvRR | RRc: 1.32 (0.97–1.81) | Not large [a] |
| Durvalumab | Feb-18 | 1 | RCT double-blind | durvalumab | placebo | 713 | Yes | PFS exA | HR: 0.52 (0.42–0.65) | Not large |
| Osimertinib | Apr-18 | 1 | RCT double-blind | osimertinib | gefitinib or erlotinib | 556 | NM | PFS exA | HR: 0.46 (0.37–0.57) | Large |
| Nivolumab | Aug-18 | 1 | Single-arm | nivolumab | no control | 109 | NM | OvRR | RRc: 1.70 (0.64–4.57) | Not large [a] |
| Dacomitinib | Sep-18 | 1 | RCT open-label | dacomitinib | gefitinib | 452 | Yes | PFS | HR: 0.59 (0.47–0.74) | Not large |
| Pembrolizumab | Oct-18 | 1 | RCT double-blind | pembrolizumab + carboplatin and paclitaxel or nab-paclitaxel | placebo +carboplatin and paclitaxel or nab-paclitaxel | 559 | Yes | OS | HR 0.64 (0.49–0.85) | Not large |
| Lorlatinib | Nov-18 | 1 | Single-arm | lorlatinib | no control | 215 | NM | OvRR | RRc: 0.90 (0.72–1.14) | Not large [a] |
| Atezolizumab | Dec-18 | 1 | RCT open-label | atezolizumab + bevacizumab, paclitaxel, and carboplatin OR atezolizumab + paclitaxel, and carboplatin | bevacizumab, paclitaxel, and carboplatin | 800 | Yes | OS | HR: 0.78 (0.64–0.96) | Not large |

(*Continued*)

**Table 2.** (Continued)

| Drug name (generic) | Approval date | N. st | Study design | Intervention arm | Control arm | Sample size | OS reported? | Primary endpoint | Effect size calculated (95%CI) | Magnitude of effect |
|---|---|---|---|---|---|---|---|---|---|---|
| Atezolizumab | Dec-18 | 1 | RCT open-label | atezolizumab | docetaxel | 850 | Yes | OS | HR: 0.74 (0.63–0.87) | Not large |

HR: hazard ratio, NM: not mentioned, OS: overall survival, OvRR: overall response rate, ORR: objective response rate, PFS: progressive-free survival, PFS exA: progressive-free survival with explanatory analysis, RCT: randomized controlled trial, RRc: risk ratio calculated.

[a] Not large and crossed the null effect.

between surrogate endpoints and overall survival with respect to lung cancer, although some studies showed a strong correlation, mostly they were weakly correlated [20].

The accelerated pathway was created in 1992 and 25 years later has been described as a "successful program" for oncology drugs with only 5% of drugs withdrawn from market due to failure to demonstrate clinical benefit. The median time for benefit to be verified and conversion from accelerated to regular approval is 3.4 years [21]. 85% of the accelerated approvals in the last 25 years were based on response rate outcomes and 72% were based on single-arm studies [21]. Considering all FDA approvals between 2005 and 2012, 448 pivotal trials were considered, of which 89.3% were RCTs and 45.3% used surrogate endpoints [12]. Similarly, in the present study, 67% of lung cancer approvals were based on surrogate endpoints and only 57% were informed by RCT evidence.

In Evidence-based medicine, core outcome measures are essential for good evidence quality [14]. Core outcomes must be patient-centered and clinically relevant. Surrogate endpoints in cancer research have been described as measurable, but not clinically meaningful [22]. It is common that a decrease in tumor size or delayed tumor growth is not related to survival gain. While not all surrogate endpoints fail to predict patient-centered outcomes, with a notable example being disease-free survival in colon cancer [23], we question the unbridled use of surrogate endpoints to grant market approval. Much has been discussed regarding value-based healthcare and the importance of a patient-centered approach and add their important healthcare results for reimbursement and payment proposals [24].

**Table 3. Characteristics of studies used to support FDA lung cancer approvals (2006 to 2018) made via the accelerated vs regular pathways and BTD vs non-BTD.**

| Study characteristics | Accelerated vs regular pathways | | | | Breakthrough therapy designation (BTD) vs non-BTD | | | |
|---|---|---|---|---|---|---|---|---|
| | Accelerated pathway n (%) | Regular pathway n (%) | Prevalence ratio (95%CI) | p-value | BTD (%) | Non-BTD (%) | Prevalence ratio (95%CI) | p-value |
| **Study design** | | | | | | | | |
| Single-arm | 11 (85) | 5 (21) | 4.06 (1.80–9.16) | < .001 | 10 (62) | 4 (25) | 2.50 (0.99–6.33) | 0.073 |
| RCT | 2 (15) | 19 (79) | | | 6 (38) | 12 (75) | | |
| **Outcome type** | | | | | | | | |
| Surrogate endpoint | 12 (92) | 14 (58) | 1.58 (1.09–2.30) | 0.057 | 13 (81) | 10 (62) | 1.30 (0.83–2.30) | 0.43 |
| Overall survival | 1 (8) | 10 (42) | | | 3 (19) | 6 (38) | | |
| **Sample size** | | | | | | | | |
| <200 | 9 (69) | 7 (29) | 2.37 (1.15–4.88) | 0.036 | 9 (56) | 9 (56) | 2.37 (0.54–1.84) | 1.0 |
| ≥200 | 4 (31) | 17 (71) | | | 7 (44) | 7 (44) | | |
| **Effect size** | | | | | | | | |
| Large | 5 (38) | 6 (25) | 1.54 (0.58–4.08) | 0.27 | 7 (44) | 2 (13) | 3.50 (0.85–14.34) | 0.11 |
| Not large | 8 (62) | 18 (75) | | | 9 (56) | 14 (87) | | |

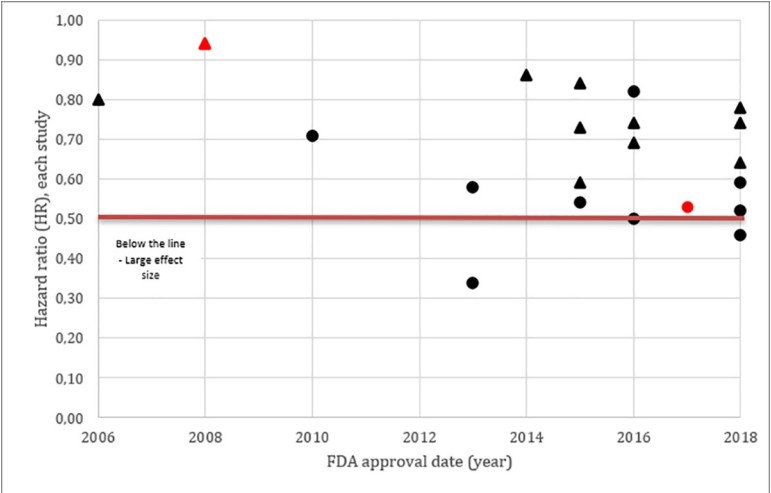

**Graph 1. Accelerated approval (red), Scatter plot distribution of each study effect sizes measured with hazard ratio as primary endpoint.** Studies using Overall Survival were represented with the triangle and Studies using Progressive-Free Survival were represented with the Circle. Color RED referred to accelerated approval pathway and the BLACK referred to regular pathway.

RCTs remain the gold standard for intervention studies and non-controlled studies can only be justified if a dramatic effect is seen [25]. Breakthrough approval is intended for drugs that are expected to provide "substantial improvement" in a "clinically significant endpoint over available therapy" [9]. Single-arm studies are case series and they carry many uncertainties because the study is not controlled, with uncontrolled patient selection and reporting bias. With no control group, it is difficult to assess effect size properly and impossible to draw causal conclusions [26]. The FDA allows phase 1 or phase 2 trials for breakthrough designation. It is encouraging that 37.5% (6 of 16) of BTDs granted for lung cancer were RCTs. Nonetheless, predicting clinical benefit using single-arm trials that measure surrogate endpoints has proven

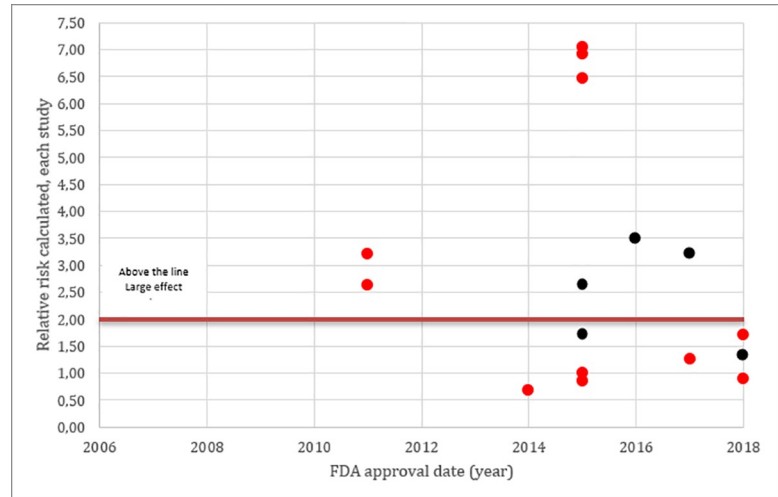

**Graph 2. Accelerated approval (red), Scatter plot distribution of each study effect sizes measured with risk ratio calculated from studies using response rate as primary endpoint.** All Studies (circle) were single arms, response rate was the primary endpoint and we calculated the risk ratio estimated by using historical control. Color RED referred to accelerated approval pathway and the BLACK referred to regular pathway.

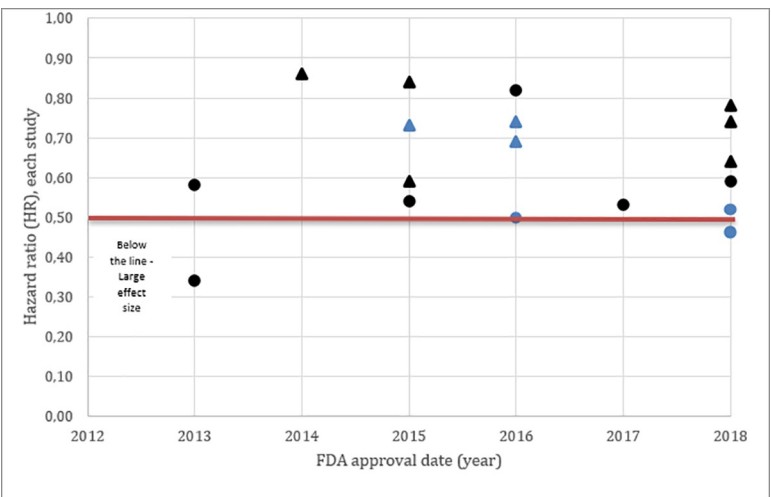

**Graph 3. Breakthrough therapy designation (blue), Scatter plot distribution of each study effect sizes measured with hazard ratio as primary endpoint.** Studies using Overall Survival were represented with the Triangle and Studies using Progressive-Free Survival were represented with the Circle. Color BLUE referred to Breakthrough therapy designation and the BLACK referred to Non- Breakthrough therapy designation.

challenging and we recommend revised BTD criteria that align more clearly with BTD's mission.

Patient pressure for new drugs for unmet medical need and market pressure for innovation are used to justify approval of drugs with more uncertainties. However, we found only a small proportion of drugs showing evidence of a large effect size. In general, studies supporting cancer drugs approved by the FDA demonstrate low to modest response rates [27, 28].

A study published in September 2019 assessed the effect size of all drugs and devices that received breakthrough therapy designation since 2012 and were based on nonrandomized

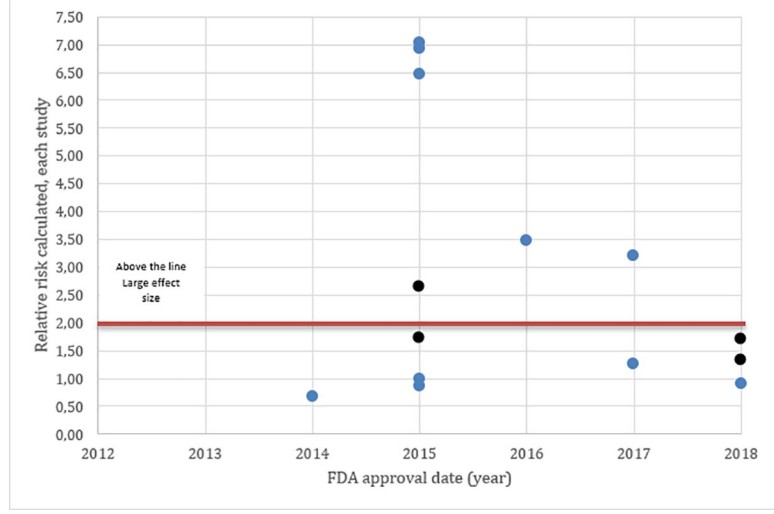

**Graph 4. Breakthrough therapy designation (blue), Scatter plot distribution of each study effect sizes measured with risk ratio calculated from studies using response rate as primary endpoint.** All Studies (circle) were single-arms, response rate was the primary endpoint and we calculated the risk ratio estimated by using historical control. Color BLUE referred to Breakthrough therapy designation and the BLACK referred to Non- Breakthrough therapy designation.

clinical trials [29]. The authors concluded that 26% of studies in which the FDA did not request further RCTs showed a risk ratio of five or greater. For drugs, it was noted that effect size was larger among studies without RCT requirements. It is notable that the FDA did not properly assess the control group data, which is crucial for effect size estimation in single-arm studies and there is no standardized threshold for effect size [29]. Despite this, there is no consensus regarding how large a truly dramatic effect size is. Some authors consider that a risk ratio equal to or larger than 5 to 10 can be safely attributed to the intervention, as it is unlikely to be solely explained by bias or confounding [30]. In our study, we took a conservative approach, considering effect sizes of two or greater as large, this was based on the GRADE recommendation. However, despite this more conservative approach, larger effect sizes were not observed in the accelerated pathway.

The argument for expedited approval is that although these studies have significant limitations, they demonstrate large effect sizes. This was not supported in our study. Similarly, elsewhere it has been shown that approvals granted BTD between from 2012 to 2017 were approved 1.9 years earlier than those did not grant this designation. However, there was no difference in the PFS or RRs for solid tumors. In our study, we found that only 44% of the drugs granted BTD had large effect sizes, and the majority demonstrated smaller effects.

## Limitations

Our sample of approvals is small and included only lung cancer treatments. However, lung cancer approvals serve as a suitable case study due to the large quantity of approvals for novel treatments in oncology (targeted drugs and immunotherapies).

We used historical controls in the estimation of effect size in single-arm studies. Although we specified transparently our process of control selection, the choice of historical controls will always be somewhat subjective and could be questioned. Historic controls likely introduce inaccuracies because: (1) the population included in the groups is different, (2) the period when the intervention was tested is different, (3) the available health care technologies and facilities may be different [31]. Moreover, we selected the historical control based on the information available to us at the time of writing. However, in reality the FDA may not have had access to this same information because the approvals processes can be concomitant.

We did not assess the bias in the included studies. This can clearly influence the certainty of the evidence. A big challenge is using a tool to assess both RCTs and single-arm studies by the same metrics.

## Conclusion

Our results show that effect size was not different between accelerated and regular pathway approvals, despite accelerated approvals being based on more single-arm studies, small sample sizes and surrogate primary endpoints, all of which are related to low evidence quality.

Surrogate endpoints were also more frequent in the accelerated pathway, as expected due to the pathway criteria, however many studies in the regular approval employed surrogate endpoints. Larger effect sizes were more frequent in relative risk estimated from studies based in response rate (single-arms).

More than half the BTD approvals were based on "not large" effect sizes, despite the definition for this designation being demonstration of a dramatic effect size.

## Supporting information

**S1 Table. Details of approvals exclusion and justification.**
(DOCX)

**S2 Table. Details of the control selection and risk ratio calculation.**
(DOCX)

## Author Contributions

**Conceptualization:** Tatiane Bomfim Ribeiro, Moacyr Roberto Cuce Nobre.

**Data curation:** Tatiane Bomfim Ribeiro.

**Formal analysis:** Tatiane Bomfim Ribeiro, Lewis Buss, Cole Wayant.

**Methodology:** Tatiane Bomfim Ribeiro, Lewis Buss, Moacyr Roberto Cuce Nobre.

**Software:** Lewis Buss.

**Supervision:** Moacyr Roberto Cuce Nobre.

**Writing – original draft:** Tatiane Bomfim Ribeiro.

**Writing – review & editing:** Tatiane Bomfim Ribeiro, Lewis Buss, Cole Wayant.

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
