## [Decision Letter · Decision Letter 0]

21 May 2020

PONE-D-20-00807

Comparison of FDA Accelerated vs Regular Pathway Approvals for Lung Cancer Treatments between 2006 and 2018

PLOS ONE

Dear Dr. Bomfim Ribeiro,

Thank you for submitting your manuscript to PLOS ONE. After careful consideration, we feel that it has merit but does not fully meet PLOS ONE’s publication criteria as it currently stands. Therefore, we invite you to submit a revised version of the manuscript that addresses the points raised during the review process.

Specifically some methodological issues were raised considering the statistical power of the studies discussed within the paper that should be addressed.

Please submit your revised manuscript within 30 days. If you will need more time than this to complete your revisions, please reply to this message or contact the journal office at plosone@plos.org. Please include the following items when submitting your revised manuscript:

We look forward to receiving your revised manuscript.

Kind regards,

Erika Cecchin

Academic Editor

PLOS ONE

Reviewers' comments:

Reviewer's Responses to Questions

**Comments to the Author**

1. Is the manuscript technically sound, and do the data support the conclusions?

Reviewer #1: Yes

Reviewer #2: Partly

2. Has the statistical analysis been performed appropriately and rigorously? 

Reviewer #1: I Don't Know

Reviewer #2: No

3. Have the authors made all data underlying the findings in their manuscript fully available?

Reviewer #1: Yes

Reviewer #2: Yes

4. Is the manuscript presented in an intelligible fashion and written in standard English?

Reviewer #1: Yes

Reviewer #2: Yes

5. Review Comments to the Author

Reviewer #1: Authors have compared FDA accelerated vs regular pathways approvals and breakthrough therapy designations for lung cancer treatment between 2006 and 2018. The authors concluded that fater approvals are in the majority full of uncertainties which should be viewed with caution and the patient have to be communicated to allow shared decision making.

The manuscript is very well written and informative. It could be accepted for publication.

Reviewer #2: Ribeiro and co. compared the approved accelerated and approved regular clinical studies. They stated that the “larger effect size” is not different from the accelerated one and the regular approval for FDA lung cancer approvals between 2006 and 2018.

This suggestion is very important, but this statement should be better critically analysed. In particular the final conclusion could be biased, as also suggested by the authors, from the surrogated end points used in regular and accelerated approved studies. The Authors reported the GRADE method, but this method is based on the quality of evidences and substantial end point , i.e. the overall survival. Even if the Authors have evidenced several limitations of the analyses ( surrogate markers, lack of control group,…) they still conclude that there are not differences in the “larger effect size”.

Specific comments

• The effect size should be better explained and an explicative table, reporting statistics for all the parameter considered, could be useful

• With regard to the study design, have the statistic power and the sample size been considered?

6. PLOS authors have the option to publish the peer review history of their article (what does this mean?). If published, this will include your full peer review and any attached files.

Reviewer #1: No

Reviewer #2: No

---

## [Author Response · Author response to Decision Letter 0]

9 Jun 2020

Dear Editor, Dear Reviewers,

Thank for your all useful comments on our manuscript. We have addressed them in the current version and we believe the manuscript has been improved. All changes made to the original version are highlighted in a marked up copy of our manuscript.

Please find our responses below.

Yours sincerely,

Tatiane Ribeiro

Reviewer #1: Authors have compared FDA accelerated vs regular pathways approvals and breakthrough therapy designations for lung cancer treatment between 2006 and 2018. The authors concluded that fater approvals are in the majority full of uncertainties which should be viewed with caution and the patient have to be communicated to allow shared decision making.

The manuscript is very well written and informative. It could be accepted for publication.

Thanks. Nothing to add.

Reviewer #2: Ribeiro and co. compared the approved accelerated and approved regular clinical studies. They stated that the “larger effect size” is not different from the accelerated one and the regular approval for FDA lung cancer approvals between 2006 and 2018.

This suggestion is very important, but this statement should be better critically analysed.

In particular the final conclusion could be biased, as also suggested by the authors, from the surrogated end points used in regular and accelerated approved studies. The Authors reported the GRADE method, but this method is based on the quality of evidences and substantial end point , i.e. the overall survival. Even if the Authors have evidenced several limitations of the analyses ( surrogate markers, lack of control group,…) they still conclude that there are not differences in the “larger effect size”.

Thanks for the comments, we clarified these aspects in the conclusion.

Specific comments

• The effect size should be better explained and an explicative table, reporting statistics for all the parameter considered, could be useful

We are grateful for the reviewer’s comments, we already information regarding table 1, table 2 and supplemental material that explain raw data. In order to add information regarding effect sizes, according to reviewers suggestion we opted to create Graphs (Graph 1-4) to demonstrate effect sizes differences in Accelerated approval pathway and Breakthrough therapy designation.

• With regard to the study design, have the statistic power and the sample size been considered?

We included this information at the Methods session: “The studies were primarily descriptive, and we included all data available for lung cancer treatment in the period (2006-2018).”

Thus, as we included all the information available of studies used to approval for lung cancer treatment, power calculation is not necessary.

---

## [Decision Letter · Decision Letter 1]

7 Jul 2020

Comparison of FDA Accelerated vs Regular Pathway Approvals for Lung Cancer Treatments between 2006 and 2018

PONE-D-20-00807R1

Dear Dr. Bomfim Ribeiro,

We’re pleased to inform you that your manuscript has been judged scientifically suitable for publication and will be formally accepted for publication once it meets all outstanding technical requirements.

Kind regards,

Erika Cecchin

Academic Editor

PLOS ONE

Additional Editor Comments (optional):

Reviewers' comments:

Reviewer's Responses to Questions

**Comments to the Author**

1. If the authors have adequately addressed your comments raised in a previous round of review and you feel that this manuscript is now acceptable for publication, you may indicate that here to bypass the “Comments to the Author” section, enter your conflict of interest statement in the “Confidential to Editor” section, and submit your "Accept" recommendation.

Reviewer #1: All comments have been addressed

Reviewer #2: All comments have been addressed

2. Is the manuscript technically sound, and do the data support the conclusions?

Reviewer #1: Yes

Reviewer #2: Yes

3. Has the statistical analysis been performed appropriately and rigorously? 

Reviewer #1: Yes

Reviewer #2: No

4. Have the authors made all data underlying the findings in their manuscript fully available?

Reviewer #1: (No Response)

Reviewer #2: Yes

5. Is the manuscript presented in an intelligible fashion and written in standard English?

Reviewer #1: (No Response)

Reviewer #2: Yes

6. Review Comments to the Author

Reviewer #1: (No Response)

Reviewer #2: NO comments. The Authors answered the questions....................................................................

............................................................................................................

7. PLOS authors have the option to publish the peer review history of their article (what does this mean?). If published, this will include your full peer review and any attached files.

Reviewer #1: No

Reviewer #2: No

---

## [Editor Report · Acceptance letter]

15 Jul 2020

PONE-D-20-00807R1 

Comparison of FDA Accelerated vs Regular Pathway Approvals for Lung Cancer Treatments between 2006 and 2018 

Dear Dr. Ribeiro:

I'm pleased to inform you that your manuscript has been deemed suitable for publication in PLOS ONE. Congratulations! Your manuscript is now with our production department. 

Kind regards, 

on behalf of

Dr. Erika Cecchin 

Academic Editor

PLOS ONE